# The Ecology of Unsheltered Homelessness: Environmental and Social-Network Predictors of Well-Being among an Unsheltered Homeless Population

**DOI:** 10.3390/ijerph18147328

**Published:** 2021-07-08

**Authors:** Mary-Catherine Anderson, Ashley Hazel, Jessica M. Perkins, Zack W. Almquist

**Affiliations:** 1Department of Earth System Science, Stanford University, Stanford, CA 94305, USA; mahazel@stanford.edu; 2Department of Human and Organizational Development, Peabody College, Vanderbilt University, Nashville, TN 37203, USA; jessica.m.perkins@vanderbilt.edu; 3Department of Sociology, University of Washington, Seattle, WA 98195, USA; zalmquist@uw.edu

**Keywords:** homelessness, unsheltered homeless populations, environmental exposure, general health, emotional well-being, social networks, gender, climate change impacts

## Abstract

People experiencing homelessness (PEH) face extreme weather exposure and limited social support. However, few studies have empirically assessed biophysical and social drivers of health outcomes among unsheltered PEH. Social network, health, and outdoor exposure data were collected from a convenience sample of unsheltered PEH (*n* = 246) in Nashville, TN, from August 2018–June 2019. Using multivariate fixed-effects linear regression models, we examined associations between biophysical and social environments and self-reported general health and emotional well-being. We found that study participants reported the lowest general health scores during winter months—Nashville’s coldest season. We also found a positive association between the number of nights participants spent indoors during the previous week and general health. Participants who spent even one night indoors during the past week had 1.8-point higher general health scores than participants who spent zero nights indoors (*p* < 0.01). Additionally, participants who experienced a conflict with a social contact in the past 30 days had lower emotional well-being scores than participants who experienced no conflict. Finally, women had worse general health and emotional well-being than men. Ecologically framed research about health and well-being among PEH is critically needed, especially as climate change threatens to increase the danger of many homeless environments.

## 1. Introduction

Compared with the general population, people experiencing homelessness (PEH) display disproportionately negative health outcomes, including higher rates of addiction, mental and physical health problems, and premature mortality [1,2,3,4,5,6,7]. However, most published research on health among PEH is limited to sheltered homeless populations. Less is known about the health of over 225,000 people experiencing unsheltered homelessness in America, who made up nearly 40% of America’s total homeless population in 2020 [8]. In this paper, we explore how biophysical and social aspects of the homeless environment influence physical and mental well-being among unsheltered PEH.

The Department of Housing and Urban Development (HUD) defines unsheltered homelessness as residing “on the street, in abandoned buildings, or in other places not suitable for human habitation [9] (p. 1)” such as in tents, cars, or parks. Although data are limited, past work finds that unsheltered PEH generally have poorer physical and mental health outcomes than sheltered PEH. For example, they have been observed to be more likely to engage in risky behaviors, such as substance use and having multiple sexual partners, and to experience higher rates of victimization and premature mortality [10,11,12]. Unsheltered PEH also face high levels of environmental exposure with limited to no protection from the biophysical environment. This situation creates a vicious cycle. Prolonged exposure to harsh environments may exacerbate the poor health of unsheltered PEH. At the same time, certain health vulnerabilities (e.g., addiction) may increase the likelihood of a person choosing to remain outdoors in harsh environments. This cycle may further increase the health risks associated with environmental exposure among an already highly medically vulnerable population.

The human-environment interactions among unsheltered PEH are multifaceted. However, collecting rich data on unsheltered PEH is difficult. Unlike sheltered PEH who can more easily be recruited at shelters, unsheltered PEH are difficult to locate and recruit because of the ways they navigate their biophysical and social environments to survive. First, unsheltered PEH may intentionally hide themselves from view due to local NIMBY (“not-in-my-backyard”) ordinances that aim to curtail the numbers of visible PEH in public spaces, sometimes even imposing criminal charges on PEH who are seen loitering or sleeping within view of public buildings and roads [13,14,15,16]. Second, many unsheltered PEH engage in illegal “shadow work”, such as sex work and selling drugs, to procure resources for themselves [14,17]. As such, they may strive to stay hidden from service providers to avoid encounters with law enforcement. Moreover, homeless advocates and researchers contend that unsheltered PEH are disproportionately under-represented in official homeless counts compared with other homeless subgroups [18,19,20]. To survive, many unsheltered PEH are intentionally concealed in tents and cars and are out of view when the HUD annual point-in-time (PIT) count is conducted on a single night in January. 

Given the abundance of health risks faced by PEH, researchers have highlighted the importance of taking an ecological perspective to studying homelessness to understand how various dimensions of the homeless environment jointly impact well-being [21,22]. In viewing homelessness through an ecological lens, “the goal is to clarify the person-environment transactions between individuals and multiple levels of the social context” and explore how these interactions may affect wellness [22] (p. 1208). In this paper, we examine how the homeless environment may be associated with both general health and emotional well-being among unsheltered PEH through physical, tangible elements of the homeless environment—such as seasonality and environmental exposure—and through the social ties that comprise the social environment. To date, no previous work has examined how both biophysical and social dimensions of the homeless environment may be jointly or differentially associated with health among unsheltered PEH in Nashville, TN.

### 1.1. Biophysical Dimensions of the Homeless Environment

Perhaps the most distinct feature of the homeless environment is the high level of environmental exposure experienced by PEH. Weather-related deaths are among the leading causes of mortality among homeless populations worldwide [23,24,25]. While extreme heat remains a serious health risk to PEH in certain geographic areas, such as in the American Southwest [26], the risks of hospital admission, poor health, and death are greater with cold weather for PEH [27,28]. Moreover, unsheltered PEH do not receive a reprieve from the outdoors by sleeping indoors overnight, even in extreme temperatures. Among 2016 PEH in 2020 in Nashville, 586 were experiencing unsheltered homelessness on a single night in January [8] where the mean daily minimum temperature was 32 °F [29]. The reasons why PEH choose to forego shelters that are theoretically available to them are usually due to structural and psychological barriers. Structural barriers to shelter access include shelters requiring unmarried heterosexual couples to sleep separately, not allowing pets, having strict curfews, and inconvenient shelter locations, to name a few [30,31,32]. Psychological barriers such as addiction, mental illness, fear of stigma and judgment, and trauma from past incarceration or institutionalization may also keep some PEH from feeling safe in shelters [10,33,34,35,36]. 

Climate change adds additional urgency to exploring the relationship between biophysical environment and health among PEH. Global, climate-induced migration is expected to force more people into refugee status in the coming years, which will likely increase the size of homeless populations globally [37,38,39]. Additionally, extreme weather events—including heat waves, wildfires, storms, and floods—are projected to increase and intensify due to climate change, leading to hazardous environmental conditions (e.g., poor air quality) that will cause even more people to lose their homes and increase the vulnerability of an unsheltered life [37,40]. These climate-related events will likely exacerbate the already precarious health of homeless populations [41,42].

Unsheltered PEH therefore represent one of the most medically and environmentally vulnerable populations in the developed world. Yet, no prior work has empirically assessed associations between biophysical features of the homeless environment—such as temperature, seasonality, precipitation, and levels of exposure—and self-reported general health and emotional well-being health among unsheltered PEH. In this study, we couple data collected from 246 unsheltered PEH in Nashville, TN, from August 2018–June 2019 with daily temperature and precipitation data from the National Oceanic and Atmospheric Administration (NOAA) Global Historical Climatology Network (GHCN) dataset. We predict that PEH will report poorer general health and emotional well-being on days that they experience harsh environmental conditions, including cold weather and prolonged environmental exposure. We also predict a positive association between general health emotional well-being and the level of social support within one’s social network, which we detail in the following sub-section. 

### 1.2. Social Dimensions of the Homeless Environment

While the ways in which the biophysical environment may affect health among unsheltered PEH are fairly obvious, less conspicuous are the ways in which social aspects of the homeless environment may affect health among PEH—such as through the influence of one’s social network. A social network is defined as a collective of entities called nodes or vertices (e.g., PEH individuals) and social relationships (e.g., the social support ties between PEH individuals) [43]. Among PEH, social networks aid in providing resources critical for survival, including emotional, financial, and material supports [14,44]. In fact, within the homelessness literature, obtaining resources from one’s social network is commonly regarded as an ecological adaption, or “coping strategy”, to the homeless environment [17,45,46,47].

Once a person becomes homeless, they may gradually replace social ties to housed friends and family with ties to other PEH. PEH social networks have been observed to vary in structure and function across numerous PEH subgroups—such as ethnic groups, gender groups, and mentally ill and substance-addicted groups [48,49]. While these networks may help facilitate survival among PEH, they may also serve as a double-edged sword in simultaneously producing negative health consequences. For example, PEH may engage in risky behaviors with their social network contacts, such as unprotected sex, drug use, and alcohol use [50,51]. The nature of relationships between PEH may also be antagonistic, wherein the stressors of homelessness combined with underlying mental and physical health problems may lead to conflict or violence, causing additional emotional burden. Additionally, extensive social ties to other PEH may contribute to a person becoming resigned to a state of chronic homelessness, or “entrenched”, in homelessness [52,53].

A large body of work details the associations between people’s social networks and health-related outcomes and behaviors—including disease spread, physical and mental health outcomes, substance use, and health norms [54,55,56,57]. Echoing this work, social networks have also been found to be associated with a variety of health outcomes and behaviors among PEH. For instance, Almquist (2020) demonstrated the importance of PEH social networks for public health outcomes through extrapolative simulation analysis of information transmission (e.g., handwashing or mask wearing) [58]. Additionally, PEH with more social support (e.g., emotional, material, and financial supports) in their social networks tend have been observed to display better emotional health, and report fewer depressive symptoms, decreased odds of suicide ideation, and increased prosocial behavior [59,60,61]. Social support has also been observed to be positively associated with physical health, decreased odds of victimization, and the utilization of health and homeless services [12,60]. Still, a majority of empirical work examining associations between social support and health among PEH has been limited to sheltered homeless populations.

Yet, several qualitative, ethnographic works suggest social networks may operate in unsheltered homeless populations similarly to those in sheltered populations [62]. For example, street-dwellers in Austin, TX, displayed the tendency to share what few resources they had with other PEH in need—including money, food, and cigarettes [17]. Additionally, a two-year ethnographic study of homeless women in downtown Atlanta found that women offered various types of social support to one other that positively impacted people’s sense of self-esteem and sense of belonging [63]. Given the lack of quantitative work examining associations between social support networks and health among unsheltered PEH, our work contributes to this literature by statistically testing for associations between features of the social dimension of the homeless environment—including various social network metrics of positive and negative social ties within one’s social network—and general health and emotional well-being. Moreover, by examining well-being among PEH using an ecological lens, we can also investigate how these dimensions may jointly or differentially affect health among unsheltered PEH.

Currently, the extent to which biophysical and social dimensions of the unsheltered homeless environment are associated with health is unknown. In this paper, we evaluate this relationship among an unsheltered homeless population in Nashville, TN. We found that participants who were surveyed during winter (i.e., Nashville’s coldest season) displayed poorer general health than participants surveyed during fall, spring, and summer. However, we did not find seasonality to be associated with emotional well-being. Additionally, while we observed no association between social support (i.e., emotional, material, and financial support) and self-reported general health or emotional well-being, we observed a negative association between the number of antagonistic ties in one’s social network and self-reported emotional well-being, suggesting that having adverse interactions with social contacts has measurable impact on emotional well-being among unsheltered PEH. By viewing unsheltered homelessness as an ecology wherein people interact with both biophysical and social dimensions of their environments, we show that biophysical aspects of the homeless environment are associated with general health, while social dimensions of the homeless environment, especially antagonistic social network ties, are associated with emotional well-being. 

## 2. Materials and Methods

### 2.1. Nashville Unsheltered PEH Data

Data on 246 PEH were collected from August 2018–June 2019 in Nashville, TN, via convenience sampling at four brick-and-mortar service facilities and three street locales where homeless services, such as mobile meals and shower trucks, were available once a week. Participants were asked if they would like to participate in a research study about health among PEH in Nashville, TN. After obtaining verbal consent, we administered a survey to participants to collect sociodemographic information—including age, gender, ethnicity, education level, LGBTQI+ status, veteran status, lifetime homelessness duration, and whether or not they had a caseworker.

To collect social network data, we employed three name generators [64,65,66,67]—a social network analysis technique to elicit social network contacts—wherein participants identified (1) family members and (2) friends they spent the most time with during the past six months and (3) the people they would go to if they needed emotional, material, or financial support. Additional information was collected on each named social contact, including their housing status, whether they had ever used drugs or alcohol with the participant, whether participants trust that social contact, and whether that social contact had ever made the participant feel upset in the past 30 days. For each named social contact in name generators 1, 2, and 3, participants were also asked if that social contact was someone they could ask for emotional, material, and financial support. 

We collected information on self-reported general health and emotional well-being, with the general health scale and emotional well-being scale within the RAND 36-Item Short Form Health Survey (SF-36) [68]. Specifically, we used SF-36 items 1, 33, 34, 35, and 36 to calculate participants’ scores for the general health scale and items 24, 25, 26, 28, and 30 to calculate participants’ scores for the emotional well-being scale. These items correspond with the general health and emotional well-being scales from the SF-36. The general health and emotional well-being scales from the SF-36 are presented in Appendix A. These scales’ scores range from 0–100 and were calculated per the RAND SF-36 scoring instructions. 

We also asked participants if a doctor or nurse had ever told them they had any of the following chronic medical conditions: diabetes, anemia, cancer, high blood pressure, heart problems, experienced a stroke, lung problems (excluding asthma), asthma, a mental health diagnosis, liver problems, epilepsy, mobility problems, osteoporosis, kidney problems, dental problems, eye problems (excluding vision), a disability, hepatitis, or HIV. To collect information on alcohol abuse, we administered the Rapid Alcohol Problems Screen (RAPS4) to participants [69]. To collect information on drug abuse, we employed an abbreviated, 5-item version of the 10-item Drug Abuse (DAST-10) [70]. We excluded those DAST-10 questions pertaining to past trauma and family due to the sensitive nature of the questions and the high vulnerability of our study population. Per DAST-10 scoring instructions, answering “yes” to only one question is enough to suggest potential problems relating to drug use. As such, it is possible to capture evidence of potential problems relating to drug use using only five questions. Following survey completion, participants were awarded a $15 gift card as compensation for their time.

In any investigation of unsheltered homeless populations, it is important to note there is no practical definition of unsheltered homelessness as a function of time spent outdoors relative to time spent in a shelter. Despite the HUD definition of unsheltered homelessness, there is no standardized measure of how long a person must reside in these locales to be classified as unsheltered. Currently, PEH are categorized as sheltered or unsheltered based solely on where they are residing only on the night of HUD’s annual PIT count. Thus, where they reside any other day of the year is overlooked in the count. In our study, we designed a sleeping locale inventory to ascertain whether, in the past 30 days, each study participant was predominantly sheltered or unsheltered. We asked whether the participant had slept at a family or friend’s house, in a car, outside in a tent, outside not in a tent, at a shelter, in a hospital, in an abandoned building, in jail, or in a hotel, and then asked participants to estimate the number of days out of the past 30 days they slept in each locale. Among our study population, only six PEH stayed in a shelter for more than half the nights during the past 30 days. Of those, only one participant had not spent any nights during the past week outdoors. However, this participant reported staying with family and friends and renting a hotel room—sleeping arrangements not counted in the PIT count. Thus, as indicated by our sleeping locale inventory, most of our study participants would fall into the technically defined category of “unsheltered” on any given night, or in some cases, be missing from the PIT count entirely.

### 2.2. NOAA Global Historical Climatology Network (GHCN) Data

We obtained daily temperature and precipitation data from NOAA GHCN-Daily database for Nashville, TN, Station ID GHCND:USW00013897 using the ‘rnoaa’ package in R [71]. We collected weather data encompassing daily minimum temperature (°C), daily maximum temperature (°C), and daily precipitation (mm) for each day in the survey period (4 August 2018 to 10 June 2019). We defined season cutoffs as follows. Summer: 4 August 2018–30 September 2018; Fall: 1 October 2018–30 December 2018; Winter: 1 January 2019–30 March 2019; Spring: 1 April 2019–10 June 2019. These weather data were then joined with our survey dataset by date. For each participant, the daily minimum temperature (°C), daily maximum temperature (°C), and amount daily precipitation (mm) were matched to the date the participant was interviewed.

### 2.3. Statistical Models

We ran a series of bivariate linear regression models (we fit this model using a least-squares dummy variable model using the R statistical computing environment) [72] to examine associations between participant sociodemographic characteristics, substance use, chronic health conditions, features of the biophysical environment (i.e., temperature, precipitation, and seasonality), and social network characteristics and self-reported general health and emotional well-being (Appendix A). All factors that were statistically significant (*p* ≤ 0.05) in the bivariate linear regressions were included in the multivariate models. One participant was removed from the analyses who had incomplete social network data. We conducted all analyses using the R statistical computing environment [73]. Additionally, given the well-documented gender differences in social network composition between men and women [73,74,75], we also conducted two-tailed, independent samples *t*-tests to examine whether social network characteristics differed between men and women in our study population of unsheltered PEH.

To examine factors associated with general health, we fit four fixed-effects linear regression models with covariates using forward model selection criterion based on improvements in the adjusted R^2^. We obtain the same results with backward and forward selection using the AIC decision criterion, which also corrects for finite populations [76]. We start with our simplest model (Model 1), including only seasonality to control for differences in self-reported general health during summer, fall, winter, and spring seasons. (we considered controlling for temperature, but Chi-square tests on the correlation with general health showed no significant effect. Thus, we excluded it from our model.) In Model 2, we added participant sociodemographic characteristics, including gender and education [77]. Education was not significant in predicting general health scores and therefore we dropped this predictor from the model. In Model 3, we added the number of nights participants spent outdoors in the past week to capture environmental exposure, as well as the number of social network members causing upset to participants in the past 30 days. However, the number of social network members causing upset to the participant in the past 30 days was not associated with general health, and we therefore dropped this predictor from the model. Finally, in Model 4, we added the number of participants’ self-reported chronic health conditions, excluding a mental health diagnosis. In Model 4, we initially included a binary categorical variable indicating whether participants had a mental health diagnosis, but it was not significant and its removal resulted in improved model fit, with a final adjusted R^2^ value of 0.87. The formal model we use to construct Model 4 is presented in Equation (1)
(1)$$Y_{i}=\sum_{s\in Seasons}\beta_{s}I\left\{S=s\right\}+\sum_{j=1}^{k}\beta_{j}X_{ij}+\varepsilon_{i}$$
where $Y_{i}$ = the general health score for *i*th PEH in the sample and $X_{ij}$ represents the covariate *j* (*j* = {*gender, # of nights participant spent indoors during the past week, total number of chronic conditions*}) for PEH *i* and ∈*_i_* represents the measurement error at person *i* (in our case assumed to be homogenous); the set of *Seasons* = {*fall, winter, spring, summer*}, *I* is an indicator function with 1 if Season *S* = *s* (e.g., S = fall) and 0 otherwise; $\beta_{s}$ represents the fixed effect for season *s* and $\beta_{j}$ represents the effect for variable *j*. In this model, the total number of chronic conditions excludes a mental health diagnosis.

To examine factors associated with emotional well-being, we fit a similar series of fixed-effects linear regression models using stepwise model selection (Models 5–8). In this case we did not see any seasonal relationship. Instead, we focused on gender differences in emotional well-being, employing gender as a fixed effect in each of these models. Participants who identified as gender non-binary were excluded from these analyses due to their small sample size. In Model 5, we fit our baseline model which only included gender (male/female). Next, in Model 6, we add social network factors, including the number of perceived financial supports, the number of trusted network members, the number of housed social network members, and the number of social network members who had caused upset to the participant in the past 30 days. We found only the number of social network members who had caused upset to the participant in the past 30 days to be statistically significant in our model. In Model 7, we added health factors, including whether the participant abused drugs and/or alcohol, and whether they had a mental health diagnosis. Additionally, we included the number of self-reported chronic health conditions, excluding a mental health diagnosis. Surprisingly, we did not find drug abuse or the number of chronic health conditions to be statistically significant. Finally, in Model 8, we removed the statistically insignificant parameters leaving us with a reduced form final model with a final adjusted $R^{2}$ value of 0.87. The formal model we use to construct Model 8 is presented in Equation (2)
(2)$$Y_{i}=\sum_{g\in Gender}\beta_{g}I\left\{G=g\right\}+\sum_{j=1}^{k}\beta_{j}X_{ij}+\varepsilon_{i},$$
where $Y_{i}$ represents the emotional well-being score for *i*th PEH in the sample and $X_{ij}$ represents the covariate *j* (*j* = {# of social network members who upset the participant in the past 30 days, alcohol abuse, mental health diagnosis}) for PEH *i* and ∈*_i_* represents the measurement error at person *i* (in our case assumed to be homogenous); the set of *Gender* = {*female, male*}, *I* is an indicator function with 1 if Gender *G* = *g* (e.g., S = female) and 0 otherwise; $\beta_{s}$ represents the fixed effect for gender *g* and $\beta_{j}$ represents the effect for variable *j*. In this model, alcohol abuse and mental health diagnosis are coded as binary categorical variables (yes/no).

## 3. Results

### 3.1. Characteristics of Sample Population

The distribution of study factors across our sample population is presented in Table 1. The mean age of our study participants was 44.9 years (SD = 10.3). Most identified as men (*n* = 167 [67.8%]), and most identified as white (*n* = 154 [62.6%]). Twenty-three of our study participants were veterans (9.3%). On average, participants spent only 1.6 nights indoors (SD = 2.2) during the preceding week.

Importantly, our study population differs somewhat from the 2019 HUD PIT count for Nashville-Davidson County, which took place during our data collection period. Per the 2019 PIT count, 585 people were experiencing unsheltered homelessness on a single night in January in Nashville-Davidson County. Of these, 121 were women (20.6%), 462 were men (79.0%), and 2 people identified as non-binary (0.3%). Additionally, 379 (64.7%) of these people were white, while 206 (35.3%) were non-white. However, demographic proportions represented in these counts vary from year to year. Table 2 displays two-tailed proportion z-score comparisons between our study population and Nashville-Davidson County PIT counts from 2016–2020 [8,78,79,80,81]. The proportion of white and non-white people in our sample was statistically indistinguishable from the Nashville-Davidson County PIT count for years 2016–2020. However, the proportion of males and females in our sample population differed from the PIT count estimations for all years except 2017. This difference is unsurprising, given the high level of uncertainty associated with PIT-count estimates of unsheltered homeless populations and the frequent underestimation of women in these counts [82]. As such, our cross-sectional convenience sample collected over 11 months (rather than on a single night, as is the case with the PIT count) may capture population demographics of unsheltered PEH in Nashville that are otherwise missed during the single night of the PIT count.

### 3.2. Nashville Weather Data

Table 3 displays the means and standard deviations of daily maximum temperature, daily minimum temperature, and daily precipitation stratified by survey season. During winter, Nashville has mean daily minimum and maximum temperatures of 1.0 °C and 12.9 °C, respectively, while during summer, Nashville has mean daily minimum and maximum temperatures of 20.8 °C and 30.4 °C, respectively. These seasonal temperature ranges in Nashville are consistent with what is expected in temperate climates with seasonal variability—that is, there is significant variance in weather variables across seasons. As such, it is possible to examine whether health varies with respect to seasonality, temperature, and precipitation. Scatterplots of the raw data for daily maximum temperatures, daily minimum temperatures, and daily precipitation on all days during the data collection period 4 August 2018, to 10 June 2019, in Nashville, TN, are presented in Appendix B Figure A1.

### 3.3. Sample Population Social Network Characteristics

An overview of participants’ social network characteristics stratified by gender (male/female) is shown in Table 4. The average social network size was 5.0 people (SD = 2.7). On average, study participants perceived they had 4.2 social network members (SD = 2.6) who could provide emotional support, 4.2 social network members (SD = 2.6) who could provide material support, and 3.4 social network members (SD = 2.5) who could provide financial support. Participants reported being able to trust an average of 4.3 social network members (SD = 2.6), and that, on average, 1.1 social network members (SD = 1.4) had upset the participant in the last 30 days. Our findings that PEH rely on social network members for emotional, material, and financial support is consistent with previous findings on the adaptive role of social networks in the lives of PEH [14,17]. 

We also observed gender differences with respect to several social network compositional features. We conducted two-tailed independent samples *t*-tests to examine social network differences between men and women (Appendix A, and we found that while women, on average, had more emotional, material, and financial supports in their social networks, this difference was not statistically significant. Men reported having more network members with whom they drank alcohol (M = 1.7, SD = 2.1) than women (M = 1.1, SD = 1.5; t(189.4) = 2.4, *p* < 0.05), while women reported having more social network members that upset them in the past 30 days (M = 1.4, SD = 1.4) than men (M = 0.9, SD = 2.3; t(124.9) = −2.3, *p* < 0.05). Women also had more unhoused social network members (M = 2.1, SD = 1.6) than men (M = 1.6, SD = 1.7; t(155.6) = −2.5, *p* < 0.05). 

### 3.4. Sample Population Health Characteristics

Overall, 145 (59.9%) of study participants had been diagnosed by a doctor or nurse with a mental health condition, 34.1% reported a high blood pressure diagnosis, 30.9% reported having asthma, 9.8% reported having diabetes, 15.2% reported having hepatitis, and 2.1% reported having HIV (Table 5). Many of these conditions occurred as co-morbidities. The distribution of the number of self-reported health conditions for participants is presented in Figure 1. Study participants had a median of 3 chronic health conditions. Additionally, 110 (44.7%) people were categorized as having problems with drinking and 109 (44.3%) people were categorized as having a problem with illegal drug use (Table 6). The distribution of general health and emotional well-being scores are shown in Figure 2. Participants displayed a mean general health score of 51.8 (SD = 27.1) and a mean emotional well-being score of 63.7 (SD = 23.6). 

### 3.5. Fixed Effects Linear Regression Models: Social and Environmental Factors Associated with SF-36 General Health Scale Score

The regression models examining associations between seasonality, sociodemographic characteristics, environmental exposure, social network factors, and chronic health conditions and SF-36 general health scores are presented in Table 7. These models produced four key findings. First, as shown in Model 1, seasonality as a fixed effect alone accounted for 79% of the variance in general health scores, suggesting a strong seasonal trend in general health among unsheltered PEH. In Model 1, the fixed-effects coefficients represent the mean SF-36 general health scores for participants who were surveyed during the summer, fall, winter, and spring, respectively. Participants surveyed during the winter displayed the lowest mean general health score (44.4), while participants surveyed during the summer displayed the highest mean general health score (57.4). Boxplots of general health scores stratified by season are presented in Figure 3. Second, we found that unsheltered women displayed considerably lower general health scores than unsheltered men. In our final model (Model 4), we observed that being male was associated with an average 8.6-point increase in general health score (*p* ≤ 0.001). Third, the number of self-reported chronic health conditions was negatively associated with general health and reporting an additional chronic health condition was associated with an average 6.2-point decrease in general health score (*p* ≤ 0.001). Fourth, Model 4 suggests that spending just a single night inside during the past week displays a significantly positive association with general health among unsheltered PEH. Participants who spent just one night indoors during the past week reported general health scores that were, on average, 1.8-points higher than participants who spent no nights indoors during the past week (*p* ≤ 0.01). 

### 3.6. Fixed Effects Linear Regression Models: Social and Environmental Factors Associated with SF-36 Emotional Well-Being Scale Score

Our four models examining associations between sociodemographic characteristics, social network factors, and chronic health conditions and SF-36 general health scores are presented in Table 8. These models produced three key findings. First, as indicated in Model 5, gender alone accounted for 88% of the variance in emotional well-being scores. In this model, unsheltered women displayed a mean emotional well-being score of 59.0 while men displayed a mean emotional well-being score of 65.9 (*p* ≤ 0.001). Second, we found that the number of social network members who upset participants within the last 30 days was negatively associated with emotional well-being score. According to Model 8, participants who had one social network member who had caused them to be upset in the previous 30 days reported emotional well-being scores that were, on average, 2.3 points lower than those participants who did not report this type of negative social contact. Finally, we found that alcohol abuse, rather than drug abuse, was highly negatively associated with emotional well-being score. In Model 8, participants who were categorized as having potential alcohol addiction per the DAST-10 displayed emotional health scores that were, on average, 10.1 points lower than participants who did not report alcohol addiction. We interpret this finding to suggest that alcohol use is a better general proxy for adverse effects of substance abuse on emotional well-being than the drug abuse question, not that drug abuse is not also negatively associated with emotional well-being. That is, while only alcohol abuse was significant in the model, it is likely that because drug and alcohol abuse are correlated, the alcohol use question provides more information on the effects of substance use on emotional well-being.

## 4. Discussion 

### 4.1. Biophysical and Social Dimensions of the Unsheltered Homeless Environment

We found that our study population displayed a mean general health score of 51.8 (SD = 27.1) and a mean emotional well-being sore of 63.7 (SD = 23.6) on a 100-point scale as measured by the SF-36 health survey. To contextualize these scores, it is helpful to compare them to the Medical Outcomes Study (MOS)—the benchmark study from which the SF-36 was adapted [68]. Among a cross-sectional sample of *n* = 2471 adults in three major cities, the mean general health score was 56.99 (SD = 21.1) and the mean emotional well-being score was 70.38 (SD = 22.0). Thus, our sample of unsheltered PEH had both lower general health and emotional well-being scores than those of the general population as measured by the MOS. Additionally, there was greater variance within general health and emotional well-being scores in our study population of unsheltered PEH compared to the MOS.

Hwang et al. (2009) examined associations between social support and physical and mental health among PEH by employing the SF-12, an abbreviated version of the SF-36 that generates a physical and mental health score. However, because the SF-12 is no longer public domain, we did not employ it in our study and therefore cannot make direct comparisons because the physical health and mental health scales (SF-12) are calculated differently than the general health and emotional well-being scales (SF-36) we used. Nevertheless, our findings lend additional support to Hwang et al. (2009), who also found that PEH displayed lower physical and mental health scores compared with the general population in the MOS. As many PEH do not receive routine medical care and therefore lack traceable medical records, future investigations into health among homeless populations could benefit from creating and/or utilizing robust, reliable standardized self-reported health metrics. Standardizing how we measure self-reported health among PEH could aid in benchmarking this population’s baseline health status. Furthermore, a standardized set of health metrics would enable easier cross-sample comparisons and improve the accuracy of measuring the success of targeted health interventions directed toward homeless populations.

Our regression models demonstrated that biophysical dimensions of the unsheltered homeless environment—such as seasonality and environmental exposure—are strongly associated with general health. For example, using Model 4, we can see that spending 7 nights inside results in a 12.6 (7 × 1.8)-point increase in general health score. Therefore, if a person spent 7 nights indoors during the winter, their general health score (holding all else constant) would be expected to be around 71.9, which is slightly higher than the baseline health of 70.1 in the summer. We interpret this as evidence that interventions and policies aiming to curtail weather-related illness and death among PEH should ensure that temporary cold-weather shelters are available to unsheltered PEH during winter. Additionally, these results suggest that warming supplies, including clothing, tents, and propane heaters, should be readily available to PEH who stay outdoors during the cold seasons to potentially help offset the health risk associated with extreme cold.

These results also demonstrate the importance of the association between gender and general health. To illuminate this effect, we can look at a hypothetical, median unsheltered person under our best fitting general health model (Model 4). Assuming this person displays the population median number of chronic health conditions (excluding one mental health diagnosis) (median = 3) and reported the median number of nights spent indoors during the past week (median = 0), this person would have general health scores of 59.9 (summer), 56.9 (fall), 49.2 (winter), and 50.0 (spring) (SE = 3.2, 2.8, 3.2, 3.5) for men, and scores of 51.4 (summer), 48.4 (fall), 40.6 (winter), and 41.1 (spring) (SE = 3.7, 3.4, 3.7, 4.0) for women. At the median level of chronic conditions, women are expected to have a general health below 50 for all seasons but summer. 

Women represent a highly vulnerable PEH subgroup [83,84]. The finding that women are more health-vulnerable than men is consistent with past work finding that homeless women display disproportionally high mortality rates compared with the general population. For instance, a review on mortality among homeless women found that while young women tend to display lower mortality rates than young men in the general population, mortality rates among young homeless men and women (under 45 years of age) were similar. This suggests that the harsh environment of homelessness can offset the standard survival advantage of being female [85]. As homeless women have also reported higher numbers of chronic conditions than men [86], cold weather may exacerbate the already precarious health of unsheltered women to a greater extent than unsheltered men.

Gender was also significant with respect to emotional well-being. In Model 5, we observed that gender alone accounted for 88% of the variance in emotional well-being scores, wherein women displayed poorer emotional well-being than men. Our findings therefore highlight the substantial gender disparities in both physical and mental health among homeless populations. This result corroborates previously published work finding that homeless women report higher numbers of stress-related symptoms and poorer mental health than homeless men [61,86]. 

Additionally, we found that participants with more antagonistic social ties (i.e., social network members who had upset them in the past 30 days) reported lower emotional well-being scores than those participants who did not report this type of negative social contact (Model 8). This result suggests that conflict with members of one’s social network may negatively impact emotional well-being among unsheltered PEH. As women reported having more social network members (mean = 1.4, SD = 1.4) who had upset them in the past 30 days than men (mean = 0.9, SD = 2.7), this highlights women’s additional social vulnerability. We interpret this result to mean that while social network ties may indeed be advantageous in providing valuable social support (i.e., emotional, material, financial), some of these ties may simultaneously be antagonistic in causing conflict or upset, especially among women. As we found no significant associations between social support and emotional well-being, this suggests that antagonistic social ties among unsheltered PEH may actually play a stronger role in emotional well-being than positive social ties.

To interpret our final emotional well-being model (Model 8), we look at the median number of contacts who upset women study participants in the last 30 days (median = 1), and the median number of contacts who upset men study participants (median = 0) combined with the condition of no mental health diagnosis or alcohol abuse. These scenarios yield emotional well-being scores for men and women of 79.4 and 73.6, respectively. Assuming a mental health diagnosis and alcohol abuse, these scores decrease to 56.2 and 50.4, respectively. As homeless women are more likely to be victims of sexual and physical abuse, and in some cases, are more likely to die prematurely from psychoactive substance use disorders, such as alcohol abuse [1], our results once again demonstrate that unsheltered women face a riskier social environment than unsheltered men.

Unlike with general health, we did not observe any relationships between biophysical features of the homeless environment and emotional well-being among our study population. However, it is possible that our survey simply did not capture the nuance of the emotional impact of consistent exposure to extreme weather conditions. One qualitative study among homeless service providers in Australia found that many of their homeless clients experienced increases in substance use, domestic violence, and loss of tents and supplies following extreme weather events [87]. Thus, the emotional state of PEH in response to weather and seasonality may present in indirect ways not captured via the SF-36. Future work could benefit from examining longitudinal changes in mental health in response to environmental changes to test whether seasonal fluctuations exist. The importance of biophysical and social environmental factors in shaping people’s well-being underscores the utility of viewing homelessness as an ecological condition, i.e., a complex web of interacting and dynamic spatial, temporal, and social variables that influences people’s ability to survive and thrive in a patterned manner. Furthermore, an ecological view of homelessness should enable better evaluation of how policies differentially affect PEH. For example, viewing homelessness through an ecological lens may better predict future epidemic spread, outbreak hotspots, and those most vulnerable to severe illness. It could help advocates better understand PEH decision patterns and expected tradeoffs when developing or applying new outreach efforts.

### 4.2. Implications for Health Impacts Due to Climate Change Among Homeless Populations

Expanding our ecological perspective of homelessness, we also frame our findings within the broader context of climate change. Currently, the impacts of climate change on homeless populations have been explored mostly in the abstract. Ramin and Svoboda (2009) suggest that climate change-induced heat waves, floods, air pollution, and changes in the epidemiology of vector-borne diseases, such as the West Nile virus, are among the most pressing concerns for homeless populations in the coming years [43]. 

For example, consider the 2020 California wildfire season—the most severe wildfire season in state history. These wildfires resulted in hazardous amounts of smoke, and resultantly, air pollution. As California has the largest homeless population in the United States of America—with around 160,000 people experiencing homelessness in 2020—it is critical to assess how climate change-related events will affect health among homeless populations [8]. A recently published study found that increases in fine particulate matter (${\mathrm{PM}}_{2.5}$), a metric of air pollution, resulting from wildfire smoke was correlated with higher respiratory hospitalizations than increases in non-wildfire ${\mathrm{PM}}_{2.5}$ conditions in California [88]. While this study did not examine hospitalizations by housing status, unsheltered PEH face more prolonged exposure to hazardous air quality conditions than housed populations resulting in adverse health consequences. Therefore, introducing housing status into conversations about how climate change may affect health among society’s most vulnerable should be of utmost public health importance. 

Our results suggest that changes in seasonality due to climate change should also be considered in conversations concerning climate change. Climate change is expected to result in temporal changes in seasonality, shifting both the timing of the onset of seasons as well as their duration [89]. Given our finding that seasonality is associated with general health among unsheltered PEH, future work should examine how climate change-induced seasonality changes may influence the health of homeless populations. Furthermore, climate change is predicted to increase frequency and uncertainty associated with extreme weather events. This increased variance makes it more difficult to predict and plan for such events, including stretches of extreme heat or cold. As unsheltered PEH are paradoxically both hidden and yet literally on the front lines of climate change in urban environments, homeless populations should be included and prioritized in planned responses to climate change.

### 4.3. Study Limitations

Our study possesses several limitations. First, because we utilized a cross-sectional convenience sampling design, it is possible that our results are not generalizable to larger-scale unsheltered homeless populations. Alternatively, our PIT-count comparisons suggested comparable demographics between our sample population and demographics encapsulated in official annual PIT-counts (Table 2). Still, it is important to note that cross-sectional samples of PEH fall short of capturing an important, yet difficult-to-measure, dimension of the homeless environment—temporal shifts in population dynamics. Recently, a framework has been proposed that suggests homelessness should be viewed as a “’moving target,’ rather than a stable state…where individuals pass through homeless episodes with beginnings, middles, and endings… and where shifting structural conditions and social policies influence rates of homelessness [41] (p. 9). While longitudinal sampling frames are, in theory, able to capture these temporal shifts in population dynamics, they also pose significant practical challenges in studying hidden populations. For instance, among unsheltered PEH, it is extremely difficult to schedule follow-up interviews with people who are hard to track down and lack consistent access to digital communication. As such, our cross-sectional sampling frame cannot capture demographic and health trends over time. However, it is suitable to capture population-level trends in health with respect to direct metrics of the homeless environment for a given population at a point in time. Second, we did not survey study participants on every day during the study period. It is therefore possible that we missed some of the variation in self-reported general health and emotional well-being associated with weather on those days. Third, the dependent variables in our study (i.e., general health and emotional well-being) were limited to self-reported metrics. We therefore lack finer-grain, clinical and biological markers to corroborate these self-reports. Finally, it is important to note that while our study design allows us to measure relationships between social and biophysical dimensions of the homeless environment and self-reported general health and emotional well-being, this design does not allow us to test for causal links between the homeless environment and health outcomes.

## 5. Conclusions

In this study, we aimed to investigate health among unsheltered PEH through an ecological lens, wherein we explored how various dimensions of the homelessness environment may jointly or differentially affect health through direct (i.e., biophysical environment) and indirect (i.e., social network) pathways. To examine associations between these factors and general health, we used a effects regression model framework. Our best fitting model indicated that features of the homeless biophysical environment, such as seasonality and duration of environmental exposure within the past week, are associated with general health. We found that participants who were surveyed in the winter reported significantly lower general health scores than participants surveyed in in the summer. Additionally, we found that spending only one night indoors during the past week was associated with a 1.8 point increase in one’s general health score (this corresponds to 12.6 points or about a 10 percent increase if a PEH spends all seven nights indoors), suggesting that interventions aiming to curtail weather-related illness and death among PEH should ensure that cold-weather shelters are available to unsheltered PEH during winter to improve general health.

In addition, we observed that gender—as well as features of the homeless social environment, including the number of antagonistic social ties in one’s social network—were associated with emotional well-being. To examine associations between these factors and emotional well-being, we again used a fixed-effects regression framework. For example, we found that women who had social network members who had upset them in the past 30 days generally had the lowest emotional well-being. Specifically, our model demonstrated that having social conflict with just one member of one’s social network in the past 30 days decreased a participants’ emotional well-being score by an average of 2.3 points. The average woman in our sample has around five people in her social network. If she was in conflict with all of them, that would equate to a decrease in emotional well-being of around 11.5 points, which equals roughly 10% of the total possible score for emotional well-being. Additionally, we found that alcohol abuse, rather than drug abuse, was highly negatively associated with emotional well-being scores. In our final model, participants who were categorized as having potential problems with alcohol displayed emotional health scores that were, on average, 10.1 points lower who did not have issues with alcohol.

Given the paradox of vulnerabilities faced by unsheltered PEH—access to urban infrastructure, but high levels of environmental exposure; frequent, yet highly constrained mobility; and supportive, yet precariously antagonistic social network ties—future work exploring health among this population could benefit from further conceptualizing various dimensions of the homeless environment, and examining how they may differentially impact health. Climate change adds additional urgency to exploring relationships between environment and health among unsheltered PEH and homeless populations more generally. 

## Figures and Tables

**Figure 1 ijerph-18-07328-f001:**
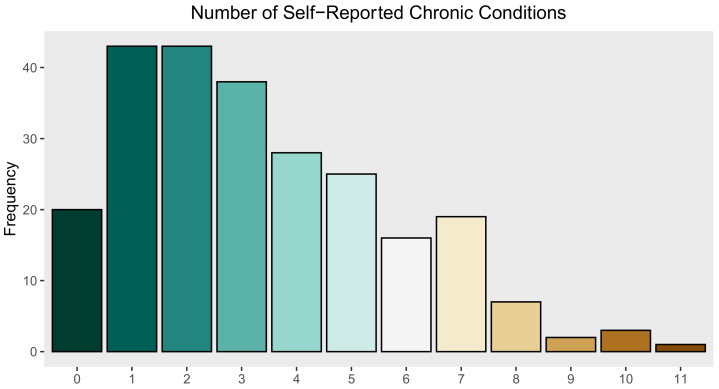
Frequency distribution of number of self-reported health conditions among *n* = 246 unsheltered PEH in Nashville, TN.

**Figure 2 ijerph-18-07328-f002:**
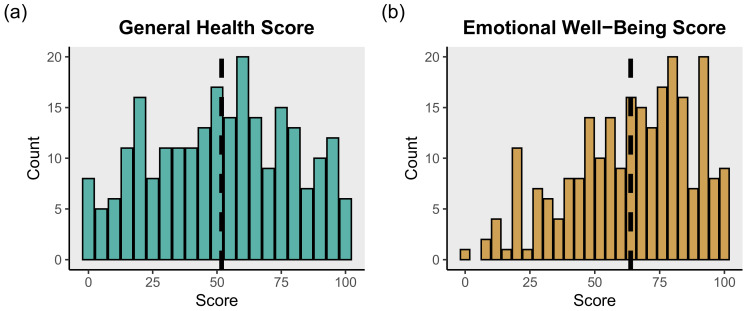
(**a**) General health and (**b**) Emotional well-being scores among *n* = 246 unsheltered PEH in Nashville, TN as measured by the SF-36. Dashed lines indicate mean scores.

**Figure 3 ijerph-18-07328-f003:**
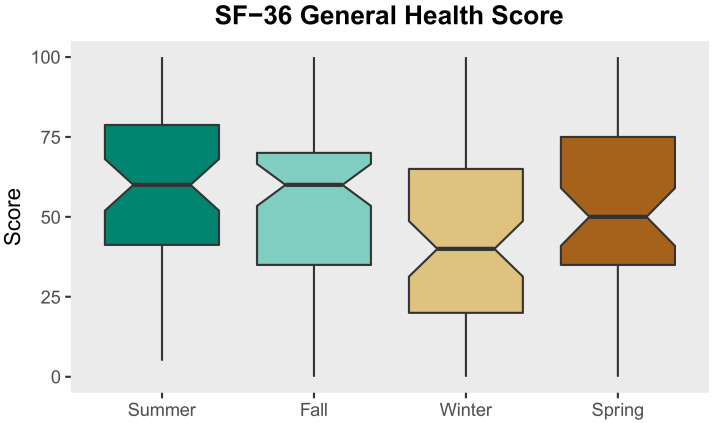
Boxplots of SF-36 general health scores stratified by survey season.

**Table 1 ijerph-18-07328-t001:** Sociodemographic characteristics of *n* = 246 unsheltered PEH in Nashville, TN.

	Men		Women		Non-Binary		Total	
	*n*	%	*n*	%	*n*	%	*n*	%
Participant Characteristics								
Gender								
Female	---	---	---	---	---	---	75	30.4%
Male	---	---	---	---	---	---	167	67.8%
Non-Binary	---	---	---	---	---	---	4	1.8%
LGBTQI+								
No	154	92.2%	67	89.3%	0	0.0%	221	89.8%
Yes	13	7.8%	8	10.7%	4	100.0%	25	10.2%
Ethnicity								
Non-White	99	59.3%	22	29.3%	2	50.0%	92	37.4%
White	68	40.7%	53	70.7%	2	50.0%	154	62.6%
Highest Level of Education								
K-11th Grade	55	32.9%	27	36.0%	3	75.0%	85	34.6%
GED or High School	75	44.9%	24	32.0%	1	25.0%	100	40.7%
Trade School or Any Higher Education	37	22.2%	24	32.0%	0		61	24.7%
Veteran								
No	147	88.0%	72	96.0%	4	100.0%	23	9.3%
Yes	20	12.0%	3	4.0%	0	0.0%	223	90.7%
Has Caseworker								
No	129	77.2%	56	74.6%	1	25.0%	186	76.0%
Yes	37	22.8%	19	25.4%	3	75.0%	59	24.0%
Lifetime Homelessness Duration								
1 year or less	31	18.6%	19	25.3%	0	0.0%	50	20.3%
1 year—5 years	50	29.9%	35	46.7%	0	0.0%	85	34.6%
5 years—10 years	48	28.7%	12	16.0%	1	25.0%	61	24.8%
10 years +	38	22.8%	9	12.0%	3	75.0%	50	20.3%
Sleeps in Encampment with Other PEH								
No	106	63.4%	40	53.3%	2	50.0%	148	60.2%
Yes	59	36.6%	35	46.7%	2	50.0%	96	39.8%
Interview Season								
Summer	37	22.2%	15	20.0%	2	50.0%	54	22.0%
Fall	46	27.5%	25	33.3%	1	25.0%	72	29.2%
Winter	47	28.1%	22	29.3%	1	25.0%	70	28.5%
Spring	37	22.2%	13	17.4%	0	0.0%	50	20.3%

	Mean	SD	Mean	SD	Mean	SD	Mean	SD
Age (Years)	46.6	10.2	41.4	9.1	39.8	16.4	44.9	10.3
Number of Nights Spent Inside During Past Week	1.6	2.2	1.5	2.2	0.8	1.5	1.5	2.2

**Table 2 ijerph-18-07328-t002:** Two-tailed proportion z-score comparison between sample demographic proportions and annual PIT count demographic proportions from 2016–2020.

		HUD Point-in-Time Count				
	Study Data *n* = 242	2016 *n* = 672	2017 *n* = 639	2018 *n* = 613	2019 *n* = 583	2020 *n* = 584
Gender						
Male	167	549 *	458	470 *	462 *	448 *
Female	75	123 *	181	143 *	121 *	136 *
		**HUD Point-in-Time Count**				
	Study Data *n* = 246	2016 *n* = 673	2017 *n* = 639	2018 *n* = 616	2019 *n* = 585	2020 *n* = 584
Ethnicity						
White	154	434	419	403	379	371
Non-white	92	239	220	213	206	213

Note: Non-binary individuals were excluded from the gender z-score comparisons, but included in the ethnicity z-score comparisons, hence the difference in sample sizes across gender and ethnicity demographics. Significance levels: * *p* < 0.05.

**Table 3 ijerph-18-07328-t003:** Mean and standard deviations of daily maximum and minimum temperatures and daily precipitation for each season of sampling period.

	Daily Max. Temp. (°C)		Daily Min. Temp. (°C)		Daily Precipitation (mm)	
	Mean	SD	Mean	SD	Mean	SD
Season						
Summer	30.4	4.1	20.8	2.1	50.9	81.8
Fall	16.7	5.8	4.8	5.6	45.4	97.6
Winter	12.9	7.9	1.0	5.5	70.1	140.0
Spring	26.0	5.2	12.6	6.2	7.6	32.8

**Table 4 ijerph-18-07328-t004:** Social network characteristics of *n* = 245 unsheltered PEH in Nashville, TN.

	Men *n* = 167		Women *n* = 74		Non-Binary *n* = 4		Total *n* = 245	
	Mean	SD	Mean	SD	Mean	SD	Mean	SD
Network Size	4.8	2.7	5.3	2.7	4.8	2.4	5.0	2.7
Number of Family in Network	1.9	1.8	2.3	2.1	1.5	1.3	2.0	1.9
Number of Friends in Network	2.5	1.9	2.7	1.9	2.8	2.2	2.6	1.9
Number of Emotional Supports in Network	4.0	2.6	4.5	2.8	4.8	2.4	4.2	2.6
Number of Material Supports in Network	4.0	2.6	4.5	2.6	4.3	2.5	4.2	2.6
Number of Financial Supports in Network	3.3	2.6	3.7	2.3	2.8	3.5	3.4	2.5
Number of Network Members with Whom Participant Uses Alcohol	1.7	2.1	1.1	1.5	1.8	2.9	1.6	2.0
Number of Network Members with Whom Participant Uses Drugs	0.9	1.5	0.7	1.0	1.5	2.4	0.9	1.4
Number of Trusted Network Members	4.3	2.6	4.4	2.8	4.5	1.9	4.3	2.6
Number of Network Members Who Upset Participant in Past 30 Days	0.9	1.3	1.4	1.4	2.0	3.4	1.1	1.4
Number of Housed Network Members	3.2	2.3	3.0	2.3	2.5	1.3	3.1	2.3
Number of Unhoused Network Members	1.6	1.7	2.1	1.6	2.0	2.8	1.7	1.7

**Table 5 ijerph-18-07328-t005:** Prevalence of self-reported chronic health conditions among *n* = 246 un-sheltered PEH in Nashville, TN.

Health Condition	*n*	% Prevalence
Diabetes	24	9.8%
Anemia	35	14.2%
Cancer	17	6.9%
High blood pressure	84	34.1%
Heart problems	31	12.6%
Stroke (has experienced)	19	7.7%
Lung problems	50	20.3%
Asthma	76	30.9%
Liver problems	24	9.8%
Epilepsy	42	17.1%
Mobility problems	72	29.3%
Osteoporosis	4	1.6%
Kidney problems	19	7.7%
Dental problems	110	44.7%
Eye problems (excluding vision)	21	8.5%
Disability	14	5.7%
Hepatitis	37	15.0%
HIV	5	2.1%
Mental health diagnosis	145	59.9%

**Table 6 ijerph-18-07328-t006:** Prevalence of substance abuse and emergency department utilization within past 12 months among *n* = 246 unsheltered PEH in Nashville, TN.

	Men		Women		Non-Binary		Total	
	*n*	%	*n*	%	*n*	%	*n*	%
Substance Use								
Alcohol Abuse								
No	89	53.4%	47	62.7%	0	0.0%	136	55.3%
Yes	78	46.6%	28	37.3%	4	100.0%	110	44.7%
Drug Abuse								
No	90	53.9%	45	60.0%	2	50.0%	137	55.7%
Yes	77	46.1%	30	40.0%	2	50.0%	109	44.3%

**Table 7 ijerph-18-07328-t007:** Fixed-effects stepwise linear regression models exploring associations between seasonality, socio-demographic factors, environmental exposure, and chronic health conditions on general health scale score (SF-36).

	Model 1		Model 2		Model 3		Model 4	
	β	95% CI	β	95% CI	β	95% CI	β	95% CI
Biophysical Environment								
Season								
Summer	57.4 ***	(50.1, 64.8)	42.8 ***	(32.6, 53.0)	49.5 ***	(40.1, 58.9)	70.1 ***	(61.5, 78.6)
Fall	53.5 ***	(47.2, 50.9)	40.1 ***	(31.1, 49.1)	44.8 ***	(36.4, 53.2)	67.1 ***	(59.0, 75.2)
Winter	44.4 ***	(37.9, 50.9)	31.5 ***	(22.6, 40.4)	33.3 ***	(24.3, 42.3)	59.3 ***	(50.4, 68.3)
Spring	53.3 ***	(45.7, 60.8)	38.5 ***	(28.2, 48.8)	41.1 ***	(30.9, 51.3)	60.1 ***	(51.1, 69.1)
Sociodemographic Factors								
Gender								
Female (reference)	---	---	---		---	---	---	---
Male	---	---	13.5 ***	(6.2, 20.8)	12.6 ***	(5.4, 19.8)	8.6 **	(2.6, 14.6)
Education								
K-11th Grade (reference)	---	---	---		---	---	---	---
GED or HS Diploma	---	---	6.2	(−1.6, 14.1)	---	---	---	---
Any Higher Education	---	---	7.7	(−1.2, 16.5)	---	---	---	---
Exposure								
Number of nights spent inside during past 7 days	---	---	---	---	2.3 **	(0.8, 3.8)	1.8 **	(0.5, 3.1)
Social Network Factors								
Number of social network members causing upset to participant in past 30 days	---	---	---	---	−2.1	(−4.6, 0.4)	---	---
Chronic Health Conditions								
Number of chronic health conditions (excluding mental health diagnosis)	---	---	---	---	---	---	−6.2 ***	(−7.4, −5.0)
Adjusted $R^{2}$	0.79		0.80		0.81		0.87	

Significance values: ** *p* ≤ 0.01, *** *p* ≤ 0.001.

**Table 8 ijerph-18-07328-t008:** Fixed-effects stepwise linear regression models exploring associations between gender, social network factors, substance use, and chronic health conditions on emotional well-being scale score (SF-36).

	Model 5		Model 6		Model 7		Model 8	
	β	95% CI	β	95% CI	β	95% CI	β	95% CI
Sociodemographic Factors								
Gender								
Female	59.0 ***	(53.7, 64.4)	56.2 ***	(49.0, 63.4)	78.4 ***	(70.8, 86.0)	75.8 ***	(68.9, 82.9)
Male	65.9 ***	(62.4, 69.5)	61.6 ***	(55.8, 67.4)	81.3 ***	(75.7, 87.0)	79.4 ***	(74.4, 84.5)
Social Network Factors								
Number of perceived financial supports	---	---	1.6	(−0.1, 3.4)	---	---	---	---
Number of trusted social network members	---	---	1.1	(−0.9, 3.0)	---	---	---	---
Number of social network members who upset participant in past 30 days	---	---	−5.3 ***	(−7.6, −3.0)	−2.4 *	(−4.7, −0.3)	−2.3 *	(−4.4, −0.1)
Number of housed network members	---	---	−0.3	(−2.3, 1.8)	---	---	---	---
Health Conditions								
Alcohol abuse								
No (reference)	---	---	---	---	---	---	---	---
Yes	---	---	---	---	−9.3 **	(−5.2, 6.5)	−10.1 ***	(−15.8, −4.4)
Drug abuse								
No (reference)	---	---	---	---	---	---	---	---
Yes	---	---	---	---	0.7	(−5.2, 6.5)	---	---
Chronic Health Conditions								
Number of Chronic Health Conditions (Excluding Mental Health Diagnosis)	---	---	---	---	−1.1	(−2.5, 0.2)	---	---
Mental Health Diagnosis								
No (reference)	---	---	---	---	---	---	---	---
Yes	---	---	---	---	−10.8 **	(−17.2, −4.4)	−13.2 ***	(−18.9, −7.4)
Adjusted $R^{2}$	0.88		0.89		0.90		0.90	

Significance values: * *p* ≤ 0.05, ** *p* ≤ 0.01, *** *p* ≤ 0.001.

## Data Availability

Data available on request due to privacy and ethical restrictions.

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
