# Peer review of "The Ecology of Unsheltered Homelessness: Environmental and Social-Network Predictors of Well-Being among an Unsheltered Homeless Population"

_ijerph, 2021, doi:10.3390/ijerph18147328_

Round 1

Reviewer 1 Report

The work presented is of enormous relevance and interest. It is worth paying attention to how environmental conditions influence the lives of homeless people.

The methodology used by the authors is novel, innovative and they arrive at extremely interesting and useful results for people who work with this group of people. Congratulations!

It would be interesting if the authors could clarify the objective of the work in the manuscript.

It would also be relevant for the authors to dedicate a paragraph to the limitations of the study.

Reviewer 2 Report

The research of this paper has an innovative perspective and complete content, and it can be published in this research field. However, the following suggestions are still put forward:

(1) It is suggested that the selection of research keywords should be more precise.

(2) At the end of the introduction, it should be the introduction to the research design of this paper, rather than the content of the research conclusion.

(3) There are also some format adjustments, such as readjusting the format of the table.

Reviewer 3 Report

In this article on the ecology of unsheltered homelessness, the authors look at health factors of street homeless individuals in terms of their environmental conditions and social networks.   Specifically, over the course of August 2018 to June 2019, they take cross-sectional, self-reported information from 246 unsheltered homeless in Nashville to analyze the influence of weather (temperature and precipitation) and the number and nature of recent social contacts on general health and emotional sense of well-being.   They find lower health in cold weather, mediated by having spent at least one night indoors during the week.   They also find that emotional well-being is diminished by those who experience conflict within the past 30 days, with women experiencing worse physical and emotional health.   

Overall, this is a very interesting article that is well-conceived, well-researched, and well-written.  Measures of physical and emotional health use established instruments.  The links between environmental factors, physical health, and potential implications of climate change are provocative, but perhaps a bit alarmist.   One does not need to know about negative impacts on homeless people in order to act on climate change, nor do we need to have climate change as a threat to get us to act on dealing with the health of unsheltered homeless people.  Notwithstanding, harsh climate conditions are a great threat to the unsheltered homeless.

My comments are few.   The authors do a good job in their language to note factors where they find an association between the homeless environment and general health and emotional well-being (and they provide a good discussion of the limitations of their cross-sectional design, and the challenges of having a longitudinal study with this population).    And yet, at times they offer policy recommendations that assume a causal pathway -- namely of environmental factors leading to the health and emotional outcomes.

Because the population is a “moving target,” we can’t rule out that population shifts account for the changes.  Nor can we rule out alternative explanations.  I’m not familiar with the dynamics of unsheltered homeless in Nashville, but it is possible that the most hardened, chronic homeless resist moving to a warmer place, or indoors during winters -- which would leave the most unhealthy outdoors.   Or, perhaps the most unhealthy do take advantage of the emergency shelter beds made available during winter months, and thus the findings of this article greatly under-estimate the impact of weather on health.   Likewise, if the most savvy at seeking resources are the ones who get the winter beds (i.e., the strongest among the group), then the finding that spending a night indoors increases health could be a selection issue.   The person who is sensitive and with poor emotional health may also be the one who is most likely to report conflict with their network.   

For this article, it is not necessarily important to ascribe causal pathways in order to act.   If those who are unsheltered experience increased suffering during the winter, then it is enough to warrant intervention (as most cities do).   If women who have poor emotional health also have high levels of reported conflict within the past month, that is worthy of attention.

Specific comments:

  • Lines 79-80.  “may jointly or differentially affect health among unsheltered PEH”...   Affect may be too strong a word, since there is no controlled manipulation of a variable with a rigorous comparison group.
  • Lines 88-89.   Good comparison of the PIT to the full year convenience sample.  
  • Lines 115-117.   It is reasonable that harsher weather be associated with poorer health, but did you consider whether there would be a lag effect?
  • Lines 256-258.   Any distinction between rain and snow in precipitation records?  These are quite different in terms of impact on unsheltered homeless people.
  • Line 311.   The R-square score is very high.   Is there multicollinearity?
  • Line 343.  I’m not sure the cross-sectional design is necessarily more accurate than the PIT count, but maybe more complete.   It is likely that the cross sectional method catches some the PIT doesn’t, and misses some the PIT counts.
  • Sect 3.3  On first reading, I wanted to see a comparison to the gen. Popn., or a poor population -- but that is offered later.
  • Line 418.   Since this is not a controlled experiment, we cannot ascribe causal links.  It is possible the association between staying indoors one night is spurious -- due to selection, for example.
  • Lines 482-485.   Cold weather shelters are generally made available to the unsheltered homeless during winter months.   Again, this research design cannot establish that making more beds available will improve measured health among the street homeless.  (Making heaters and tents available could lead to more people on the streets and poorer health.)
  • Lines 503-507.  In the field of homelessness, it is commonly thought that single, adult females who are homeless, especially unsheltered, are a particularly vulnerable group -- not because they are more deeply affected by the harsh environment, but because perhaps women in our society are expected to seek help and receive help.   Perhaps, those women who end up on the street have really come to the end of their rope.   But certainly, cold weather would exacerbate these conditions.
  • Lines 518-519.  Or, women with negative emotional well-being report more conflict in their relationships (which ties in with the previous comment about who becomes street homeless).  
  • Lines 541-542.  Speculative.  One can think of other explanations.
  • Line 559-583.   Good discussion of limitations.   You may want to acknowledge that the design limits you to identifying associations, and does not establish causal links.
  • Lines 640-643.   Any evidence that the relationship between number of conflicts and emotional well-being is linear?  Did you try alternate specifications? 
